# Handedness in Animals and Plants

**DOI:** 10.3390/biology13070502

**Published:** 2024-07-05

**Authors:** Silvia Guerra, Umberto Castiello, Bianca Bonato, Marco Dadda

**Affiliations:** Department of General Psychology (DPG), University of Padova, 35131 Padova, Italy; umberto.castiello@unipd.it (U.C.); bianca.bonato@unipd.it (B.B.); marco.dadda@unipd.it (M.D.)

**Keywords:** handedness, climbing plants, lateralisation, asymmetry, chirality, animals, humans, kinematics, comparative biology

## Abstract

**Simple Summary:**

Animals and plants present asymmetric structures in nature. The most relevant motor behavioural manifestation of lateralisation is handedness, which is defined as the consistent use of one effector rather than the other in performing certain tasks. In animals, including human beings, handedness is associated with the presence of a nervous system. Researchers have recently challenged this idea by reporting that even organisms without a nervous system, such as plants, exhibit similarities with animals in terms of directional movement patterns (i.e., right-handed prevalence), opening up the possibility of a comparative study of handedness across taxa. Here, we advance a comparative approach to the study of handedness in plants by adopting the experimental paradigms already used to research laterality in various animal species.

**Abstract:**

Structural and functional asymmetries are traceable in every form of life, and some lateralities are homologous. Functionally speaking, the division of labour between the two halves of the brain is a basic characteristic of the nervous system that arose even before the appearance of vertebrates. The most well-known expression of this specialisation in humans is hand dominance, also known as handedness. Even if hand/limb/paw dominance is far more commonly associated with the presence of a nervous system, it is also observed in its own form in aneural organisms, such as plants. To date, little is known regarding the possible functional significance of this dominance in plants, and many questions remain open (among them, whether it reflects a generalised behavioural asymmetry). Here, we propose a comparative approach to the study of handedness, including plants, by taking advantage of the experimental models and paradigms already used to study laterality in humans and various animal species. By taking this approach, we aim to enrich our knowledge of the concept of handedness across natural kingdoms.

## 1. The Asymmetric Nature of the Universe


*“Life as manifested to us is a function of the asymmetry of the universe and of the consequences of this fact”.*
Pasteur, 1860

Nature is asymmetric at all levels, and most of the elements characterizing it have a structure that differs from their mirror image, a phenomenon termed chirality. Chirality is present from the simplest form of an element, such as molecules. A molecule exists in two mirror image forms that cannot be superimposed (i.e., enantiomers). Mirror-image molecules could be either Levorotatory enantiomers (i.e., L-enantiomers) or Dextrorotatory enantiomers (i.e., D-enantiomers), and they are produced in equal amounts (i.e., homochirality) [1]. These molecules are crucial for the development of specific proteins for the structure and chemical regulation of living cells and their DNA (i.e., deoxyribonucleic acid), and both exhibit a fixed chirality (e.g., DNA presents a right-handed helix form) [1,2]. Cells mostly comprise chiral molecules, which define the cell’s left–right (L-R) asymmetric morphology. Cell chirality is observed in protozoans (i.e., groups of unicellular eukaryotes); eukaryotic cells, such as metazoan (i.e., specialised cells in tissues and organs that have a division of labour and perform specific functions) [1]; the blastomeres (i.e., a type of cell produced by cell division of the zygote after fertilisation) of various invertebrate species [3]; and vertebrate cultured cells [4,5]. Recent evidence has suggested that the intrinsic cell chirality underlies the L-R asymmetric morphogenesis in invertebrates (e.g., drosophila) [6,7] and vertebrates (e.g., zebrafish melanophores) [8]. From this perspective, this mechanism may contribute to the asymmetric development of L-R asymmetry across phyla [1].

From plants to human beings, all organisms present a structure that is not identical to its mirrored image. Variation in the anatomical structure (i.e., position and orientation of organs) across and among species (e.g., plants and animals) is often described in relation to the primary organism’s axes [2]. For example, animals with frontally placed eyes have large binocular overlap and combine images from the two eyes into one percept whereas animals with laterally placed eyes have two separated visual fields, and this might lead to a different analysis of visual stimuli [9]. Animal species, including human beings, present three whole-body axes, as follows (Figure 1A,B): (i) the anterior–posterior (A–P) axis, which extends longitudinally from head to tail; (ii) the dorsal–ventral (D–V) axis, in which ventral typically faces toward and dorsal away from a substrate; and (iii) the left–right (L–R) axis, which is defined in relation to a plane running along the A–P midline. In plants, the axes are related to the direction of growth of the organismal parts (e.g., stem, roots). 

The main axis of plant growth is the apical–basal (A–B), which represents a single straight line from the origin of the plant to the tip of the shoot or the roots (Figure 1C) [10,11]. Its direction (i.e., polarisation) depends on various environmental cues (e.g., gravity) [10,12], and it remains active throughout plant development by defining the growth direction of the stem and the roots [13,14]. The A–B axis is related only to the plant’s simplest growth forms. However, the stem and the root system present their own relative elongation, which is characterised by a rhythmic repetition of leaves or secondary roots, respectively, at different distances. For instance, along the stem, leaves are added asymmetrically in the form of a single leaf primordium (i.e., groups of cells that will form into new leaves) to one side of the growing point as an investment around the stem. In this case, symmetry can also be observed in the leaves, which are characterised by several axes, such as the proximal–distal (i.e., distance from the base to the apex; Figure 2A), the D–V (i.e., distance from the adaxial to the abaxial side; Figure 2B), and the medial–lateral (i.e., distance from the midvein to the margin; Figure 2C). In plants, “proximal” and “distal” should be applied to organ parts that do not develop from an apical meristem (e.g., leaflets, petals; Figure 2A) or to organs with an apical meristem that branch out from another site (e.g., branches or lateral roots). In the adaxial–abaxial axis, the adaxial part includes the top of the leaf and the abaxial the bottom of the leaf (Figure 2B). Then, the medial–lateral axis is used to describe laminar plant parts, such as many leaves and petals and some shoot axes (e.g., cactus paddles), that expand through the growth of marginal meristems (Figure 2C) [15]. 

Furthermore, symmetry can be observed in several plant organs (e.g., leaves, roots, and shoots). It can be (i) radial (i.e., similar parts are arranged in a balanced way around the centre of the plant’s body; Figure 3A), such as in the development of petals in flowers, roots, and shoots; (ii) dorsiventral (i.e., the front and back parts of the plant’s organs; Figure 3B), such as in leaves; (iii) bilateral (i.e., the left and right parts of the plant’s body are symmetrical; Figure 3C), as in the flowers of pea plants; or (iv) helical/spiral (i.e., there is a central vertical axis around which the plant’s organs, such as the stem, twist, either towards or away; Figure 3D), as in the arrangement of leaves along the stem (i.e., phyllotaxy) [16]. 

Asymmetries represent a way to adapt to the predominance of various environmental conditions. Humans, together with many other species, exhibit directionally cerebral and functional asymmetries that in some cases are linked, such as the localisation of the language centre in one hemisphere and the position of the organs (i.e., the heart is on the left, the liver is on the right side, and the small and large intestines coil in a chiral manner) [17,18]. On the other hand, in plants, the arrangement of leaves in a helix/spiral along the stem is necessary to balance the mass of the leaves along the stem and to provide an optimal structure for maximising light capture, efficient gas exchange, and protection from excessive damage by insects, wind, or the sun [16,19,20]. Functional asymmetry can also be observed at the level of motor behaviour, that is, how an organism behaves and interacts with the environment. Functional motor asymmetries can be exemplified at the level of specific effectors, such as hand/foot dominance. The most prominent and studied example of lateralisation is the hand/limb/paw dominance and performance, also known as handedness (i.e., the consistent use of one effector rather than the other in performing certain tasks) [21,22,23,24,25]. In some cases, the meaning of handedness is clear, as in hand dominance in human beings. In other cases, when bodies are characterised by irregular shapes (e.g., plants), the designation of handedness is far more critical. 

At this point, the reader may wonder how handedness can be used to study lateralisation in brainless organisms, such as plants. Plants do not have a nervous system and do not develop along the same axes as animals do. However, as animals and plants co-evolved, they heavily interacted, and it is not surprising that several features of both groups of organisms are subjected to selection pressures, such as symmetry vs. asymmetry. This opens up the possibility of a comparative taxonomic and evolutionary study of handedness focused on animals and plants. Naturally, we cannot equate brain asymmetries with those exhibited by plants. However, in terms of directional movement patterns, some factors (i.e., right-handed prevalence) can be compared across species. In the following paragraphs, we introduce the phenomenon of handedness in animals and plants. We then examine the empirical studies on handedness in separate animal species. This will lead us to propose a comparative approach to the study of handedness in plants by taking advantage of the experimental models and paradigms already used to study laterality in various animal species. Finally, to determine the basis for valid cross-species comparisons, we highlight those factors that, from our perspective, should be considered in future research.

## 2. Handedness: One Term, Multiple Meanings

Before proceeding, we must acknowledge that when species other than humans are considered, not all comparative researchers refer to the term “handedness” for phenomena commonly labelled as behavioural laterality/directional asymmetry. For example, behavioural laterality has been investigated in a large number of species, considering either parts or their entire body. The asymmetry of movement and preferred side are observed in coiling in snakes [26], tail-wagging in dogs [27], and trunk movements in elephants [28,29]. Even when limbs are present in non-human species, the literature refers to limb preference instead of handedness, underlining an anatomical–functional distinction compared to handedness in our species [30]. The determination of handedness in plants is more critical than in animals due to several factors, such as the presence of irregular body shapes and the absence of specific effectors (e.g., hands, feet). This leads to the arbitrary determination of handedness in plants from the observer’s point of view and discrepancies in the definition of right-handed and left-handed plants. Considering all this in the present perspective paper, we adopt (though with a certain degree of caution) the term “handedness” (also as a matter of consistency with the existing literature on plants), being aware that this term’s meaning is related to the species investigated.

The concept of handedness includes the idea that an organism uses one effector (i.e., a hand) more frequently than the other and the related idea that performance is more skilled or efficient with the preferred effector [2,31]. Handedness is commonly associated with the various specialisations of function that are related to the left and right hemispheres [32,33]. Until the 1980s, the idea persisted that directional asymmetry was unique to *Homo sapiens* given that right-handedness was linked to the emergence of language [34]. As commonly known, this applies to a majority of humans exhibiting a left-hemispheric dominance for the control of speech [35]. In the 1970s and early 1980s, the findings of Nottebohm [36] on laterality regarding song control in birds and those of Rogers [37] and Andrew [38] regarding visual responding in domestic chicks lay the groundwork for a series of in-depth studies on numerous other species. Today, we are aware that lateralisation is a characteristic that defines a wide range of species and that there is no discontinuity in its evolution [39,40,41]. 

Handedness is by far the most studied behavioural asymmetry in humans; in over 50 years of research, it has been studied with consideration for genetic [42,43], behavioural [44], and environmental aspects [45]. Handedness can be distinguished into two forms: left- and right-handedness. Almost 90% of humans are right-handed, and this percentage holds consistently across cultures [42,46]. Handedness varies in strength (i.e., individuals who may be weakly or strongly lateralised) and direction (i.e., left or right) within and across species. Handedness is often task-dependent (e.g., writing, reach-to-grasp, reach-to-eat), which makes identifying the factors driving the expression of handedness difficult [47,48,49]. Furthermore, handedness can be observed at the population level (i.e., when most individuals in the same population show the same bias for the right or the left) or at the individual level (i.e., a consistent preference for a single individual irrespective of the population). 

Describing a pattern of human handedness is much more complex than often reported. The methodologies used to assess handedness in humans are themselves a source of variability. The use of self-report questionnaires can introduce the risk of response bias and is affected by the motor task used (i.e., using scissors and/or unscrewing a jar), which can produce an overestimation of right bias. To investigate interspecific comparisons, some studies are based on the observation of naturalistic spontaneous behaviours [50] even though few data are available and direct observation of spontaneous behaviours is rare. Furthermore, handedness in humans seems considerably affected by cultural differences [51,52]. For instance, in Western societies, the prevalence of reported left-handers varies between 2 and 13%, whereas in other cultures, it varies from 2 to 27% [53].

Whether non-human species exhibit functional directional asymmetries comparable to human handedness is still debated, considering the extreme preference documented in our species for the right hand, a laterality which is unmatched by limb preference in other species, except for a few species of parrot [54] and ground-living kangaroos [55].

A recent review of non-human primates indicates, indeed, that population-level handedness is rare outside our species [56]. Nevertheless, evidence of consistent hand or limb preference has been documented in a host of vertebrates, including fish, rodents, birds, and anurans [57], as well as some species of prosimians [58]. Moreover, evidence of the predominance of right-handedness in captive chimpanzees for specific complex tasks (e.g., bimanual feeding) has suggested a continuity between humans and our closest relative [47]. An extensive body of literature on animal species [57] has indicated how strongly brain asymmetries affect everyday behaviour regardless of the presence of hands or limbs (e.g., fish and/or reptiles). In fish, behavioural asymmetries include a preferential ventral fin use among blue gourami, *Trichogaster trichopterus* [59], and lateralised pectoral spine stridulation among catfish, *Ictalurus punctatus* [60]. Snakes exhibit a lateralised use of a hemipenis according to different temperatures [61], whereas Roth [26], while observing coiling posture in cottonmouth snakes (used for defensive and offensive strikes directed at predators and prey), found a preference for counterclockwise coiling. The above evidence suggests that the study of behavioural asymmetry is not solely limited by the presence of limbs.

## 3. Handedness in Plants

As previously reported, asymmetries represent a way to adapt to the prevailing environmental conditions at any time. In plants, a clear example of adaptability is represented by the helical growth (i.e., circumnutation) of the stem and related extensions (e.g., tendrils, leaves) [62,63]. This movement involves many behavioural patterns, such as (i) the arrangement of the leaves or petals, (ii) the twisting of the flat plant’s organs to increase their rigidity, (iii) exploratory movements of the stem or tendrils (i.e., filamentary organs sensitive to contact and used exclusively for climbing), and (iv) the coiling of tendrils around the potential support in the environment. For instance, climbing plants adopt the helical pattern of movement to find a potential support (e.g., wooden pole, tree trunk) in the environment and, once perceived, to direct their organs, such as tendrils (i.e., modified leaves that a climbing plant uses to climb a potential support), toward it for climbing [62,63,64]. Further, the direction of the tendrils’ helical growth determines the side of the coiling movement around the support, which could be clockwise (i.e., on the left side), counterclockwise (i.e., on the right side), or mixed (i.e., both directions; Figure 4A) [64,65,66,67,68,69,70]. This direction could be fixed within species or flexible between organs of the same plants. 

Many plant species exhibit a helical structure with a preferred direction (e.g., honeysuckle grows in a left-handed helix), indicating a sort of handedness or chirality (i.e., right- and/or left-handed helices of the plant’s organs during growth) [70,71,72]. Some botanists, such as Hugo von Mohl, defined right-handed plants as those that climb a support “with the sun” or “like the clock hands” [67,69]. On the other hand, botanists such as Charles Darwin claimed that when the circle made by the finger runs clockwise, the plant’s spiral is right-handed [63], and when the circle is counterclockwise, it is left-handed [67,69]. Kihara [69] adopted the terminology that described “clockwise” as “right-handed” and “counterclockwise” as “left-handed”. Hashimoto [73] suggested defining it as “left-handed” clockwise helical growth. Indeed, clockwise arrangements of petals and leaves are correlated with left-handed helical epidermal cell files, and counterclockwise arrangements are correlated with right-handed epidermal helices. To avoid misunderstandings, here, we use the Hashimoto [73] definition: everything that turns clockwise and moves away from the observer is considered left-handed (Figure 4A), and counterclockwise movement is right-handed (Figure 4A). 

Examples of handedness in plants include (i) the spiral arrangement of leaves along the stem (i.e., phyllotaxy). For instance, the coconut tree (*Cocos nucifera*) shows a clockwise (i.e., left-handed) or counterclockwise direction (i.e., right-handed) of phyllotaxy, depending on the plant’s location in the northern or southern hemisphere. Indeed, right-handed forms are predominantly present in the northern hemisphere, with left-handed forms being present in the southern hemisphere [72,74,75]. Such examples also include (ii) the heliotropic response of plants’ organs, which follows the sun during the day. Heliotropic adaptation of plant organs (e.g., leaves) with appropriate torsions (i.e., left and/or right-handed) allows plants to orient themselves at right angles to the light source to optimise light capture and improves plant performance and fitness [62,76,77]. Another example is (iii) the helical growth of various plant organs (e.g., the stem, leaves, petals). The turn of the shoot of Arabidopsis (*Arabidopsis thaliana* L.) and the Ipomea purpurea (*Ipomea purpurea* Roth) plants is usually clockwise [78], whereas bean shoots (*Phaseolus vulgaris* L.) and Fallopia baldschuanica (*Polygonum baldschuanicum* Regel) display a counterclockwise direction [79]. (iv) Examples also include the coiling of the tendrils or stem in climbing plants. Some plants, such as pea plants (*Pisum sativum* L. cv, Alaska), show a mixed pattern of coiling direction (i.e., clockwise and/or counterclockwise direction). Jaffe [68] observed that the prevalence of the tendril’s rotation in pea plants was predominantly clockwise (i.e., 53%; 47% counterclockwise). Recent studies have confirmed these findings, showing that the direction of the circumnutation in pea plants during the approach-to-clasp movement towards a potential support (i.e., a wooden pole or another plant) can be either clockwise or counterclockwise and that it can change in the same plant [80,81]. However, except for a few cases, such as among pea plants, 92% of climbers show a right-handed coil-direction preference independent of hemispheric locations, latitude, circumnutation direction, and the thigmotropic response given by the first contact of the plant’s organ (e.g., tendril) with the supporting host or stimulus [65]. These results showed that the phenomenon of twining is not random and could be genetically determined [82]. At the molecular and cellular levels, the direction of the helical movement is linked with the arrangement of the arrays of the microtubules (i.e., hollow tubes made of alpha and beta tubulin that are a part of the cell’s cytoskeleton) in the cellular cortex and the related coiling of the cellulose microfibrils around the cell. Further, a range of microtubule-associated proteins are involved in helical growth development, such as the SPIRAL1 (i.e., SKU6), SPIRAL2 (i.e., TORTIFOLIA1), and the gene GCP2. The right or left direction depends on the microtubule arrays’ structural properties. For instance, right-handed organ twisting is always associated with left-handed microtubule arrays and vice versa. In sum, genetic and cellular investigation have demonstrated a strong link between microtubule arrangement and the potential twisting of cells and tissues [70].

The above evidence suggests that the pattern of left/right handedness in plants appears to differ depending on the plant organ (i.e., stem, leaves, tendrils), species (e.g., pea plants), and development (e.g., coconut palm phyllotaxy, plant twisting on a potential support), implying that handedness in plants may have various causes [73]. Symmetry in plants is, indeed, a plastic trait which is determined by the plant’s physiological state (i.e., plant fitness) and various environmental conditions [83,84]. In this view, the handedness preference in plants may be determined by various factors (i.e., genetic, biological, and environmental factors) and may have evolved in various ways to allow plants to function more efficiently in their surroundings by increasing their chances of survival. 

## 4. Toward a Comparative Study of Handedness in Animals and Plants?

When we refer to handedness in humans, other animal species, and plants, the differences are far more obvious than the similarities. As aforementioned, handedness is a complex concept to deal with the variability in hand use in our species. Also, it is still debated whether non-human species exhibit functional forelimb asymmetries comparable to human handedness. In a potentially comparative context, it is essential to underline whether the adoption of the term “handedness” for plants refers to the presence of a preferred growth direction (i.e., circumnutation and/or twisting), and it can be summarised as that property shown by two forms of structures that are mirror images of each other. They are often distinguished by the names right (dextral) and left (sinistral), which may be arbitrarily determined [70].

Taking this into account, an increasing number of studies have suggested that plant and animal behaviours show strong similarities (e.g., decision-making, learning) and that a neural architecture is not ubiquitously necessary to support certain abilities [80,81,85,86,87,88,89,90,91,92,93,94,95,96,97]. Can this statement be extended also to the phenomenon of handedness? In other words, is it possible to enhance the concept of handedness in plants by including functional aspects as for animals? 

As reported previously, a critical point is that handedness in plants is determined by the observer’s perspective; therefore, it is quite difficult to unequivocally define plants as right- or left-handed [67,69,73]. Another critical aspect is how and when to measure handedness. Indeed, the spiral arrangement, the helical growth, and the coiling of tendrils or the stem are all used as indicators of handedness, and to our knowledge, there is no agreement on what the most reliable measure may be or on the correlation between the various measurements. This led us to another consideration: which phases of plant growth are most indicative of handedness? Given that all of the indicators are useful, which moment is the most critical to consider?

Having said that, we found evidence of what can be potentially described as handedness at the population level in plants. Ninety-two percent of climbing plants appear to be right-handed based on the coiling direction [65], but this percentage drops to almost the chance level (53%) for some other plants [68]. It is rather unclear to what extent these two outcomes are comparable and to what extent they reflect an actual asymmetry at the population level. Do the authors estimate handedness at similar phases? Do the authors share the same criteria to assess handedness? Is it possible to hypothesise a scenario in which to investigate the variability of handedness in plants? The answer is potentially yes but with great caution. Given the fact that handedness in plants can be measured in terms of direction (right vs. left) and, perhaps, in terms of extent (e.g., the timing with which the tendrils coil around the support), only systematic studies could clarify whether there is a direction of growth and climbing common to the majority of plants observed in the same conditions with a shared methodology. There is a second critical aspect that might be difficult to address as far as handedness in plants is concerned. In humans and other animal species, handedness is often measured through repeated observations. With plants, this is obviously difficult to achieve given that on many occasions, the moment at which plants climb a support represents a non-repeatable event. A possible strategy to overcome this obstacle can be the study of plants that achieve their objective through a form of trial-and-error approach. For instance, Guerra and collaborators [80] investigated the growing movement of the apex and the tendrils from the germination of the seed until the clasping of the support. The results revealed that as the number of leaves increases, the velocity and the time taken to perform a circumnutation increase. Also, this corresponds with a decrease in the number of circumnutation and “handedness” switches. In light of this, at the population level, even if there are similar proportions of plants with clockwise, counterclockwise, and mixed directions of circumnutation movement, single plants might exhibit a preference that could be revealed through the acquisition of repeated measures. 

Furthermore, how can we measure the consistency of handedness at the individual level? Even though handedness in primates is task dependent, once again, the situation is far more complex in plants. For example, in humans, manual preference is extremely strong if measured in a writing task but is greatly reduced as the task used to assess it varies. Ideally, handedness should be measured in multiple contexts and for as many functions as possible. It could be very informative to investigate handedness as a task-dependent function in plants. For instance, it might be of interest to assess how climbers’ handedness (e.g., clasping a support) and/or the goal (e.g., competition or cooperation with other plants) varies in various tasks. In other words, investigating handedness at the individual level is a key factor in estimating the consistency of lateralisation across various tasks. However, as previously reported, the above approaches limit the categorisation of the plant as right- or left-handed at the time they have completed their growth process. We are practically blind to the range of adjustments that precedes, for instance, the climbing of a support and that can enhance the analysis of lateralised behaviour in plants.

To uncover the nature of handedness in plants, we propose to characterise plants’ movement by means of three-dimensional (3D) kinematical analysis. This methodology permits researchers to define the time course of changes in the position and orientation of the body or one of its parts (e.g., effector) in terms of trajectories, velocities, and accelerations. In human beings and various animal species, this approach has been applied in studying the handedness of goal-directed motor programmes, such as the reach-to-grasp movement, at various stages of development. Evidence has shown that right-hand preference is already observed in the foetus during hand-to-face movements [98,99]. Right- and left-handed foetuses were faster in reaching targets requiring greater precision (i.e., eye and mouth) with their dominant (vs. nondominant) hand [98]. Hand preference was also observed in young infants for grasping objects (i.e., toys or block), in which movements showed straighter and smoother paths of the dominant effector and a shorter movement time compared to the nondominant one [100,101,102,103,104,105,106]. Similar findings have been observed in adults, whose movements performed with the dominant hand were generally faster, whereas grasping movements with the nondominant hand presented a wider safety margin [107,108,109,110,111,112]. In other words, movements performed with the nondominant hand showed a longer movement time and more correction adjustments than movements with the dominant hand, possibly to compensate for a greater endpoint variability (i.e., variability of finger positions on the object). These findings have also been confirmed in non-human primates [113,114,115]. Evidence has shown that chimpanzees who used a precision grip to grasp small pieces of food were more likely to use their right hand and that this hand preference may reflect a property of the brain that is ancient and hardwired [107,113,115]. 

Recently, the kinematical approach has been extended to the study of goal-directed movement in plants. Specifically, the approaching and clasping movements performed by various organs of a pea plant towards a potential support have been characterised [80,81,85,86,87,88,89,97,116]. These findings have demonstrated that pea plants are able to perceive an element in the environment and to modulate the kinematics of their movement in terms of velocity, acceleration, aperture of their tendrils, and smoothness on the basis of the feature of the to-be-grasped support [80,87,88,89,97], the task (e.g., decision-making) [81], and the context (e.g., competition or cooperation) [85]. The tracking and analysing of plants’ movement through time and space using dedicated in-house software [115] is a potential tool for studying the distribution and functional significance of laterality in plants. Also, many of the kinematical features can be considered in a lateralised fashion. Figure 5 depicts the possible kinematical features to quantify specific properties of handedness in plants.

All of these measures might be applied to investigate laterality in plants, with specific reference to climbing plants (e.g., peas and/or beans). Climbers are an ideal model that, through their development, can somewhat represent the vegetal side of manual preference. For instance, pea plants are annual climbing plants that need to find a potential support in the environment to climb and to reach the light source, which are necessary for its survival. Their leaves are arranged asymmetrically along the stem, and each leaf has tendrils and filamentary organs that allow the plant to clasp an external support (Figure 6). 

The morphological structure of pea plants will allow us to assess various aspects of handedness, such as (i) whether clasping the support occurs predominantly in a clockwise or counterclockwise direction, (ii) whether plants have a preference for climbing a support with leaves developed on their right or left side, (iii) whether this preference could be observed in terms of a difference in kinematical patterning (e.g., velocity profiles, movement time, number of changes in direction), and (iv) movements’ smoothness (e.g., number of submovements, endpoint variability). In this case, because plants do not have right and left sides, the R–L axis in plants should be determined by the observer’s perspective (Figure 6), and (iii) the movement of right- or left-handed plants changes based on the support’s position (e.g., on the left or right side with respect to the plant).

At this point, the reader may wonder whether data obtained with a methodology allowing for a comparison across species (i.e., kinematics) may help in theorising about lateralisation. Several theories have been proposed to explain the extreme effector preference in the animal kingdom [31]. The right-shift theory [33] posited that the formation of behavioural asymmetries depends on brain asymmetries, which are induced by genes (i.e., RS+). McManus [43] suggested that genes directly affect handedness but not hemispheric dominance. The lack of explanatory power of some of the central theoretical arguments has led to a reduction in the likelihood of a purely genetic inheritance of handedness and to alternative explanations. The postural origins hypothesis states that all anthropoids, regardless of their ecology (i.e., arboreal or terrestrial primate species) share a right-hand bias in object manipulation [117,118]. For humans, a feedback specificity has been suggested for each brain hemisphere. That is, the preferred hand relies on visual feedback, while the non-preferred hand on proprioception [119]. Despite these proposals, it is still not clear how the phenomenon of handedness evolved and/or why right-handedness dominates in humans and non-human primates. One of the main reasons is the lack of homogeneity across the experimental approaches [120,121]. The use of kinematics could overcome this issue by providing a standardised methodology to study the phenomenon of handedness among and across species. Remember that recent evidence suggests that kinematical specifiers used to distinguish the use of the dominant from the nondominant hand in humans [98,107,110] can now be identified in plants [116]. This may allow us to quantify ecological variables’ influence on the evolution of handedness across taxa. The discovery of a possible pattern of correspondence in handedness across species through the expression of movement could provide an integrative view of this phenomenon and a resulting theoretical understanding of its functional and evolutionary benefits. 

## 5. Conclusions

In the present review, we examined the question of laterality across kingdoms. What may emerge from our work is that even though animals and plants are different organisms with different structures, they exhibit similarities in terms of directional movement patterns (i.e., right-handed prevalence), opening up the possibility of a comparative study of handedness across taxa. However, research on plants’ handedness is still in its early stages, and the existing literature is sparse and often controversial. The term “handedness” is controversial when adopted for humans, other animals, and plants, as pointed out. We believe that from a comparative perspective, it is appropriate to refer to the term “handedness” when describing certain plants’ behaviour, being aware of the structural and functional differences that occur between the species investigated. Another crucial point is whether to consider any possible common aetiology between animals and plants regarding laterality. In humans, functional asymmetries have been described earlier than structural asymmetries, and for a long time, the idea that anatomical and functional asymmetries might be related was simply not considered [122]. The opposite is true for research on non-human animals, among whom structural and functional asymmetries (e.g., in the avian brain) have been described as interrelated [123]. In this framework, one might argue that during evolution, similar solutions (e.g., laterality) have been selected to solve similar problems in such disparate and diverse species. Here, we suggest that the use of critical and specific methods would help identify, define, and evaluate the functional aspects of handedness in plants, as widely documented in various animal species. As knowledge about plant behaviour expands, the similarities between plant and animal behaviour are becoming increasingly evident. What may emerge from the study of handedness in plants and animals is the realisation that they complement each other nicely and, if nothing else, once again demonstrate just how similar all free-living organisms are to one another.

## Figures and Tables

**Figure 1 biology-13-00502-f001:**
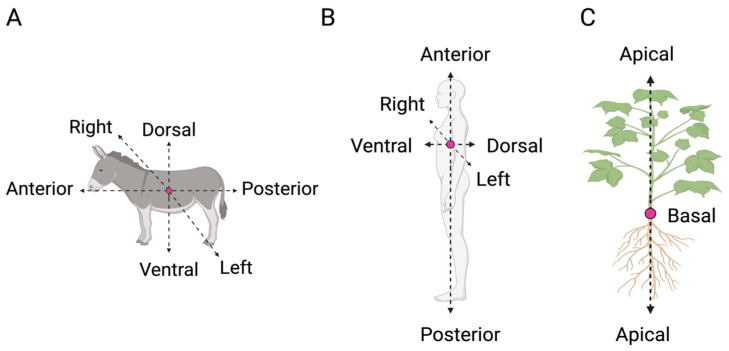
Primary axes in animal and plants. In animals (**A**) and human beings (**B**), there are three main axes: the anterior–posterior (A–P) axis (from the head to the tail/feet), the dorsal–ventral (D-V) axis (the upper or back side of an organism), and the right–left (R–L) axis (defined in relation to a plane running along the anterior–posterior midline). In plants (**C**), the main axis is the apical–basal (A–B) axis, which refers to the straight line from the origin of the plant either to the tip of the shoot or the roots.

**Figure 2 biology-13-00502-f002:**
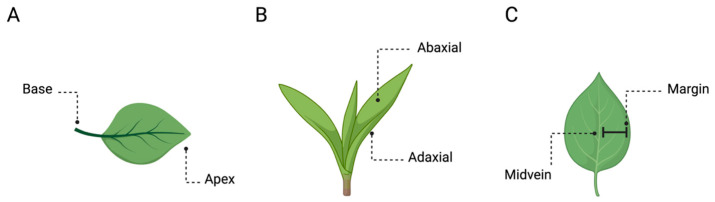
Plants’ axes. Example of (**A**) proximal–distal, (**B**) dorsal–ventral, and (**C**) medial–lateral axes in the plants’ leaves.

**Figure 3 biology-13-00502-f003:**
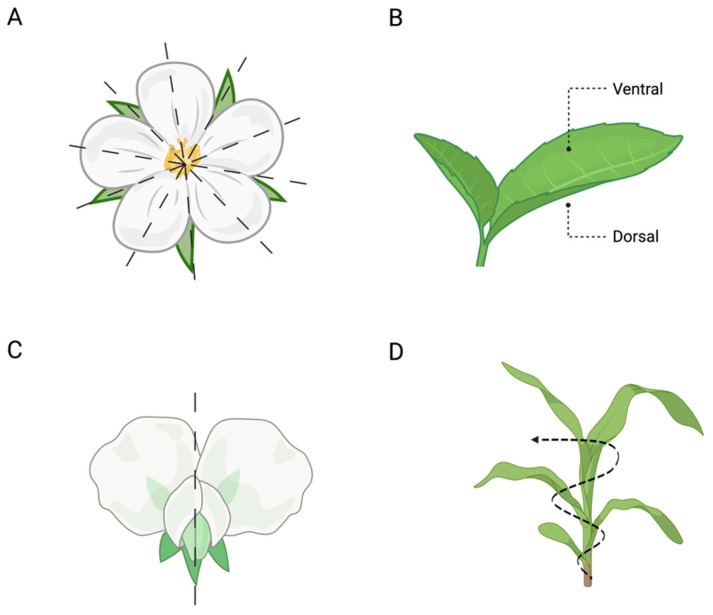
Symmetry in plants. (**A**) Example of radial symmetry in flowers (e.g., wild roses), (**B**) dorsolateral symmetry (e.g., leaves), (**C**) bilateral symmetry (e.g., flower of a pea plant), and (**D**) helical/spiral symmetry (e.g., the arrangement of the leaves along the stem).

**Figure 4 biology-13-00502-f004:**
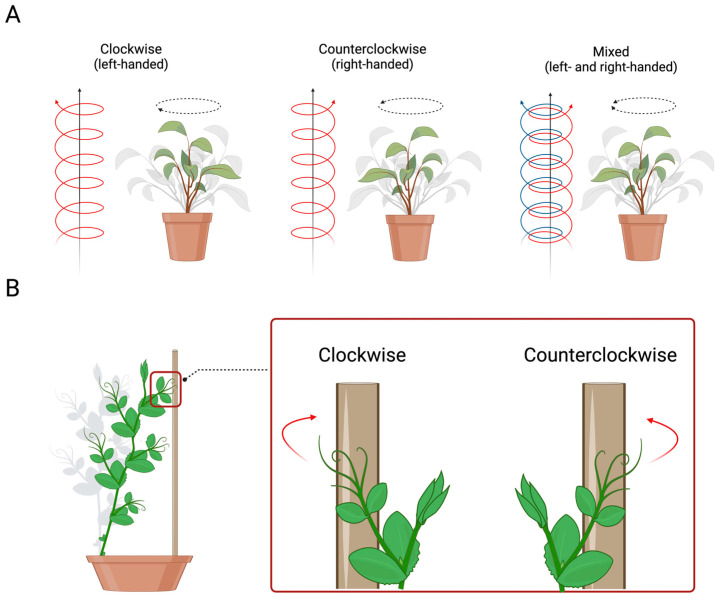
(**A**) Graphical representation of the direction of circumnutating growing movement in plants, which could be clockwise (i.e., left-handed), counterclockwise (i.e., right-handed), or mixed. (**B**) Graphical representation of the direction of the coiling movement of the tendrils in pea plants.

**Figure 5 biology-13-00502-f005:**
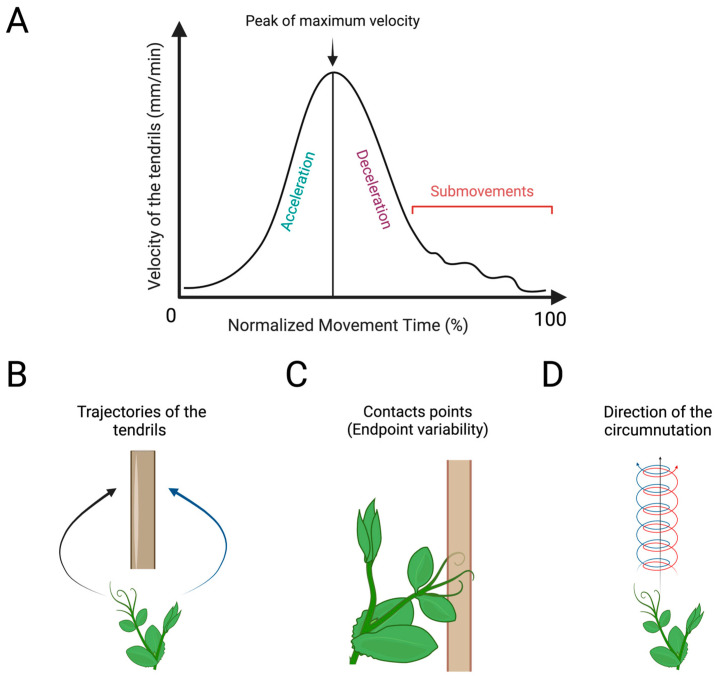
Graphical representation of the kinematical measures to quantify handedness in plants. (**A**) Graphical representation of the velocity profile of a pea plant’s tendrils during the approaching and clasping movement towards a potential support. Normalised movement time (%) refers to the time between the beginning and end of the movement. Acceleration/deceleration refers to the increasing and decreasing of the tendrils’ velocity before it clasps the support. The peak of maximum velocity (%) refers to the maximum velocity tendrils reach over the entire movement. Submovements are the corrective adjustments in the tendrils’ trajectories and velocity to execute a proper motor behaviour. (**B**) The path of the tendrils’ trajectories during the approaching and clasping of the support. (**C**) The endpoint variability refers to the variability of the tendrils’ contact point on the support. (**D**) Number of changes in direction (i.e., clockwise and/or counterclockwise) during circumnutation.

**Figure 6 biology-13-00502-f006:**
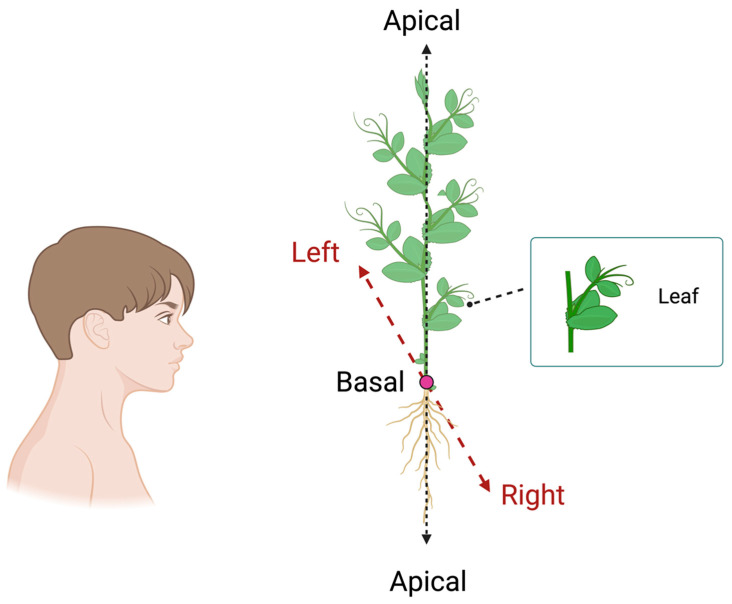
Graphical representation of the morphological structure of a pea plant’s leaves. The black dashed lines represent the apical–basal axis (A–B), and the red dashed lines represent the right–left (R–L) axis, which is determined by the observer’s perspective.

## Data Availability

Not applicable.

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
