# Peer review of "Handedness in Animals and Plants"

_biology, 2024, doi:10.3390/biology13070502_

Round 1
Reviewer 1 Report
Comments and Suggestions for Authors
I thought this review paper was very well written with excellent figures. My main concern is I disagree with the authors that the term ‘handedness’ is interchangeable with ‘laterality’ (section 2 starting on line 123). In my view, laterality is the larger organizing term for which handedness is a very well-studied example. The evidence the authors present in this section is mixed. For example, the authors in reference 25 do not call what they studied handedness but behavioral laterality/directional asymmetry. These terms are widely used by comparative laterality researchers. For example, researchers studying trunk movements in elephants or tail movements in monkeys say laterality, not handedness, although the phenomenon of handedness in humans is often the backdrop for research questions. Looking at the literature cited on snakes in references 27/28, those authors do use the term handedness to talk about different behavioral asymmetries such as coiling. My view then becomes that some researchers studying nonhuman models are using the term ‘handedness’ when they are not studying hand use but others, including myself, do not. I think the statement on lines 164-166 is overly strong and does not fairly represent the literature. My request for a revision is to acknowledge that not all comparative researchers use the term ‘handedness’ in this way.
My other comment is that while kinematics have been used in handedness studies, this tool is not the ‘go to’ for measuring handedness. It’s a much larger time investment as far as analyses go, and the results have not been as straightforward as section 4 suggests. What would improve this section is an updated literature search on handedness and kinematics across the lifespan in humans as well as nonhuman primates. Currently, there are no references given for the adult summary, and other references in this section are not up to date. I was curious how the authors thought kinematic data from plants would inform theories of lateralization that currently exist, if they could speculate on potential advancements beyond common methods.
Author Response
1. I thought this review paper was very well written with excellent figures. My main concern is I disagree with the authors that the term ‘handedness’ is interchangeable with ‘laterality’ (section 2 starting on line 123). In my view, laterality is the larger organizing term for which handedness is a very well-studied example. The evidence the authors present in this section is mixed. For example, the authors in reference 25 do not call what they studied handedness but behavioral laterality/directional asymmetry. These terms are widely used by comparative laterality researchers. For example, researchers studying trunk movements in elephants or tail movements in monkeys say laterality, not handedness, although the phenomenon of handedness in humans is often the backdrop for research questions. Looking at the literature cited on snakes in references 27/28, those authors do use the term handedness to talk about different behavioral asymmetries such as coiling. My view then becomes that some researchers studying nonhuman models are using the term ‘handedness’ when they are not studying hand use but others, including myself, do not. I think the statement on lines 164-166 is overly strong and does not fairly represent the literature. My request for a revision is to acknowledge that not all comparative researchers use the term ‘handedness’ in this way.
R1. We agree with the Reviewer regarding the possible confounding effect concerned with the terminology. Following the Reviewer suggestions, we added a paragraph (lines 146-163, p. 5; 189-199, p. 6) in which this issue is clarified. We have also removed the statement in lines 164-166.
2. My other comment is that while kinematics have been used in handedness studies, this tool is not the ‘go to’ for measuring handedness. It’s a much larger time investment as far as analyses go, and the results have not been as straightforward as section 4 suggests. What would improve this section is an updated literature search on handedness and kinematics across the lifespan in humans as well as nonhuman primates. Currently, there are no references given for the adult summary, and other references in this section are not up to date.
R2. We have now integrated this section by adding updated literature on handedness and kinematics across the lifespan in humans and non-human primates. The following references have been added:
1) Clark, L. D. E.; Iskandarani, M.; Riggs, S. L. The effect of movement direction, hand dominance, and hemispace on reaching movement kinematics in virtual reality. In Proceedings of the 2023 CHI conference on human factors in computing systems, 2023, 1-18.
2) Nelson, E. L.; Konidaris, G. D.; Berthier, N. E.; Braun, M. C.; Novak, M. F.; Suomi, S. J.; Novak, M. A. Kinematics of reaching and implications for handedness in rhesus monkey infants. Dev. Psychol., 2012, 54, 460-467. DOI: 10.1002/dev.20604
3) Mazzarella, J.; Richie, D.; Chaudhari, A. M.; Tudella, E.; Spees, C. K.; Heathcock, J. C. Task-Related Differences in End-Point Kinematics in School-Age Children with Typical Development. Behav. Sci., 2023, 13, 528. https://doi.org/10.3390/bs13070528
4) Fagard, J. The nature and nurture of human infant hand preference. Ann. N. Y. Acad. Sci., 2013, 1288, 114-123. https://doi.org/10.1111/nyas.12051
5) Gonzalez, C. L.; Whitwell, R. L.; Morrissey, B.; Ganel, T.; Goodale, M. A. Left handedness does not extend to visually guided precision grasping. Exp. Brain Res., 2007, 182, 275–279. doi:10.1007/s00221-007-1090-1.
6) Nelson, E. L.; Berthier, N. E.; Konidaris, G. D. Handedness and reach-to-place kinematics in adults: left-handers are not reversed right-handers. J. Mot. Behav., 2018, 50, 381-391. https://doi.org/10.1080/00222895.2017.1363698
7) Stins, J. F.; Kadar, E. E.; Costall, A. A kinematic analysis of hand selection in a reaching task, Laterality, 2001, 6, 347-367, DOI: 10.1080/713754421
8) Grosskopf, A.; Kuhtz-Buschbeck, J. P. Grasping with the left and right hand: a kinematic study. Exp. Brain. Res., 2006, 168, 230–240. https://doi.org/10.1007/s00221-005-0083-1
9) Castiello, U.; Dadda, M. (2019). A review and consideration on the kinematics of reach-to-grasp movements in macaque monkeys. J. Neurosci., 2019, 121, 188-204. https://doi.org/10.1152/jn.00598.2018
3. I was curious how the authors thought kinematic data from plants would inform theories of lateralization that currently exist, if they could speculate on potential advancements beyond common methods.
R3. The proposed use of kinematical methods and its possible relevance for a further understanding of laterality has been nested and discussed in the light of available theories (lines 450-474, p. 12-13). The following references have been added:
1) Marcori, A. J.; Okazaki, V. H. A. A historical, systematic review of handedness origins. Laterality, 2020, 25, 87-108. https://doi.org/10.1080/1357650X.2019.1614597
2) Annett, M. Left, right, hand and brain: The right shift theory. London: Lawrence Erlbaum Associates. 1985. doi: 10.1016/0301-0511
3) McManus, I. C. Handedness, language dominance and aphasia: A genetic model. Psychological Medicine. Monograph Supplement, 1985, 8, 3–40. doi: 10.1017/S0264180100001879
4) MacNeilage, P. F.; Studdert-Kennedy, M. G.; Lindblom, B. Primate handedness reconsidered. BBS, 1987, 10, 247–263. https://doi.org/10.1017/S0140525X00047695
5) MacNeilage, P. F. Present status of the postural origins theory In: Hopkins WD, editors. In The Evolution of Hemispheric Specialization in Primates. Oxford: Elsevier. 2007, 59–91. https://doi.org/10.1016/S1936-8526(07)05003-8
6) Goble, D. J.; Brown, S. H. The biological and behavioral basis of upper limb asymmetries in sensorimotor performance. Neurosci. Biobehav. Rev., 2008, 32, 598–610. doi: 10.1016/j.neubiorev.2007.10.006
7) Bryden, P. J. The influence of M. P. Bryden’s work on lateralization of motor skill: Is the preferred hand selected for and better at tasks requiring a high degree of skill? Laterality, 2016, 21, 312–328. doi: 10.1080/1357650X.2015.1099661
8) Palmer, R. D. Development of a differentiated handedness. Psychol. Bull., 1964, 62, 257–272. doi: 10.1037/h0046588
Reviewer 2 Report
Comments and Suggestions for Authors
This manuscript attempts to draw parallels between handedness in animals (mainly in humans), and direction of coiling in climbing plants. While this approach is creative, the paper fails to make a convincing argument about the similarity of these two manifestations of laterality. In my opinion, the differences are far more obvious than the similarities. Therefore, I suggest that the authors deal only with direction of coiling in plants and delete all the parallels that they attempt, unsuccessfully, to draw between lateral bias in plants and handedness in animals. The paper could also benefit from adding some new data on measurement of asymmetry in plants.
In its present form, the manuscript simplifies the measure of handedness and its measurement in humans and other animal species by not considering it in detail and not discussing the various methods of assessing it. It is this simplification that allows the authors to equate handedness in animals with asymmetry in plants. In other words, the overlap between laterality in animals and plants is dubious and unconvincing.
Some more minor points:
Simple Summary, line 8: Here it is stated that ‘the most relevant behavioural manifestation of lateralization is handedness”. To what is it relevant and why is it relevant? This needs to be made clear. The same issue is present in line 103.
Abstract, line 17: I very much doubt that all cases of asymmetry are homologous! Although I agree that laterality in invertebrates might have been a precursor to laterality in vertebrates, and therefore homologous, the opening of the Abstract seems to imply that laterality is homologous in all forms of life, even in plants. The opening to the Abstract should be written again, and clearly.
39: This statement is incorrect. Most vertebrate species have their eyes placed laterally.
129: “Back in the day” is a vague and rather strange phrase. It should be replaced by a statement of until when this was. In fact, it was the discoveries of laterality in songbirds, chicks and rats made in the 1970s and 1980s that disproved the belief that laterality is unique to humans.
132-133: This sentence should be supported by citing a reference, or several references.
146-148: Indeed, these attempts to show handedness in nonhuman species (primates) were unsuccessful. Here it should made clear that laterality in nonhuman species was first discovered as laterality for song control in birds and differences between the eye systems in control of behaviour in chicks. See the book “Divided Brains” by Rogers, Vallortigara and Andrew (2013) for more discussion of this.
149-150: This is incorrect. What about the footedness in some species of parrot?
166-167: What are the “no specialized functions” of plants? This is a vague statement.
211: What does “depending on the hemispheric locations” mean? It needs to be clear in this sentence even though the meaning becomes clear in the next sentence. I suggest “depending on the plant’s location in the northern or southern hemisphere”.
241-243: The reader will need to be convinced of the validity of this statement. Is time needed to perform a response the main difference? In my opinion, handedness in animals is entirely different from the “handedness” that the authors attribute to plants in this paper.
Comments on the Quality of English Language
The English is fine.
Author Response
1. This manuscript attempts to draw parallels between handedness in animals (mainly in humans), and direction of coiling in climbing plants. While this approach is creative, the paper fails to make a convincing argument about the similarity of these two manifestations of laterality. In my opinion, the differences are far more obvious than the similarities. Therefore, I suggest that the authors deal only with direction of coiling in plants and delete all the parallels that they attempt, unsuccessfully, to draw between lateral bias in plants and handedness in animals. The paper could also benefit from adding some new data on measurement of asymmetry in plants. In its present form, the manuscript simplifies the measure of handedness and its measurement in humans and other animal species by not considering it in detail and not discussing the various methods of assessing it. It is this simplification that allows the authors to equate handedness in animals with asymmetry in plants. In other words, the overlap between laterality in animals and plants is dubious and unconvincing.
R1. According to the Reviewer opinion, we have critically reworded section 2 highlighting the differences between the concept (and the use) of the term handedness between humans, animals and plants. We mentioned the difficulties in measuring handedness in humans and limb preferences in animals (see also lines 308-317, p. 9). We believe that in this revised form the manuscript does not fail in equate inappropriately handedness in such different species.
Some more minor points:
2. Simple Summary, line 8: Here it is stated that ‘the most relevant behavioural manifestation of lateralization is handedness”. To what is it relevant and why is it relevant? This needs to be made clear. The same issue is present in line 103.
R2. We have now clarified this content by defining to what and why the phenomenon of handedness is the most relevant behavioural manifestation of lateralization (lines 121-127, p. 4-5). The following references have been added:
1) Papadatou-Pastou, M.; Ntolka, E.; Schmitz, J.; Martin, M.; Munafò, M. R.; Ocklenburg, S.; Paracchini, S. Human handedness: A meta-analysis. Psychol. Bull., 2020, 146, 481. https://doi.org/10.1037/bul0000229
2) Nelson, E. L.; Karimi, A. Systematic Review: The Development of Behavioral Laterality Across the First Year of Life in Nonhuman Primates. Symmetry, 2023, 15, 1335. https://doi.org/10.3390/sym15071335
3) McManus, I. C. The history and geography of human handedness. Language lateralization and psychosis, 2009, 37-57.
4) McManus, I. C. The incidence of left-handedness: a meta-analysis, 2009.
5) Isparta, S.; Töre-Yargın, G.; Wagner, S. C.; Mundorf, A.; Cinar Kul, B.; Da Graça Pereira, G.; et al. Measuring paw preferences in dogs, cats and rats: Design requirements and innovations in methodology. Laterality, 2024, 1-37. https://doi.org/10.1080/1357650X.2024.2341459
3. Abstract, line 17: I very much doubt that all cases of asymmetry are homologous! Although I agree that laterality in invertebrates might have been a precursor to laterality in vertebrates, and therefore homologous, the opening of the Abstract seems to imply that laterality is homologous in all forms of life, even in plants. The opening to the Abstract should be written again, and clearly.
R3. We have now modified the opening of the Abstract by following the reviewer’s suggestion (lines 16-17, p. 1).
4. 39: This statement is incorrect. Most vertebrate species have their eyes placed laterally.
R4. The sentence “Animals, for instance, exhibit frontally rather than laterally placed eyes and therefore divide their perceptual world according to coordinates such as right–left, top–bottom, and front–back” has now reworded in “For example, animals with frontal rather than lateral eyes divide their perceptual world along coordinates such as right-left, top-bottom and front-back.” in the new version of the manuscript (lines 58-60, p. 2).
5. 129: “Back in the day” is a vague and rather strange phrase. It should be replaced by a statement of until when this was. In fact, it was the discoveries of laterality in songbirds, chicks and rats made in the 1970s and 1980s that disproved the belief that laterality is unique to humans.
R5. We have reworded the sentence (line 168, p. 5) in the new version of the manuscript.
6. 132-133: This sentence should be supported by citing a reference, or several references.
R6. Agree. We added the following references:
1) Boulinguez-Ambroise, G.; Aychet, J.; Pouydebat, E. Limb preference in animals: new insights into the evolution of manual laterality in hominids. Symmetry, 2022, 14, 96. https://doi.org/10.3390/sym14010096
2) Hopkins, W. D. Neuroanatomical asymmetries and handedness in chimpanzees (Pan troglodytes): a case for continuity in the evolution of hemispheric specialization. Ann. N. Y. Acad. Sci., 2013, 1288, 17-35. https://doi.org/10.1111/nyas.12109
3) Rogers, L. J. Laterality in animals. Int. J. Comp. Psychol., 1989, 3. Doi: 10.46867/C48W2Q
7. 146-148: Indeed, these attempts to show handedness in nonhuman species (primates) were unsuccessful. Here it should made clear that laterality in nonhuman species was first discovered as laterality for song control in birds and differences between the eye systems in control of behaviour in chicks. See the book “Divided Brains” by Rogers, Vallortigara and Andrew (2013) for more discussion of this.
R7. We agree with the Reviewer and we have mentioned this at p. 5-6 lines 171-174.
8. 149-150: This is incorrect. What about the footedness in some species of parrot?
R8. We have reworded the sentence (lines 200-204, p. 6) in the new version of the manuscript.The following references have been added:
1) Kaplan, G.; Rogers, L. J. Brain size associated with foot preferences in Australian parrots. Symmetry. 2021, 13, 867. doi: 10.3390/sym13050867.
2) Giljov, A.; Karenina, K.; Ingram, J.; Malashichev, Y. Parallel emergence of true handedness in the evolution of marsupials and placentals. Curr. Biol. 2015, 25, 1878–1884. doi: 10.1016/j.cub.2015.05.043.
9. 166-167: What are the “no specialized functions” of plants? This is a vague statement.
R9. We have removed the sentence in the new version of the manuscript.
10. 211: What does “depending on the hemispheric locations” mean? It needs to be clear in this sentence even though the meaning becomes clear in the next sentence. I suggest “depending on the plant’s location in the northern or southern hemisphere”.
R10. We thank the Reviewer for raising this point. As suggested, we have reworded the sentence (line 262, p. 8) in the new version of the manuscript.
11. 241-243: The reader will need to be convinced of the validity of this statement. Is time needed to perform a response the main difference? In my opinion, handedness in animals is entirely different from the “handedness” that the authors attribute to plants in this paper.
R11. We have reworded these sentences in the new version of the manuscript (lines 308-323, p. 9).
Reviewer 3 Report
Comments and Suggestions for Authors
I write this appraisal as one who has studied wide ranging examples of handedness in plants.
I found this review OK as far as it goes but extremely limited. I can’t see anything novel in it, either in terms of new perspectives or novel synthesis.
To merit publication I feel this paper needs a much more holistic approach to chirality beginning with double helix DNA molecules and then going on to consider microfibril helices in plant cell walls. Next comes the dextral and sinistral asymmetry of motile cells. If one looks specifically at mosses, peristome twisting and protonemal phototropic and thigmotropic responses immediately come to mind.
With the addition of this kind of information this paper could be of general interest but not as it stands at present.
Author Response
1. I feel this paper needs a much more holistic approach to chirality beginning with double helix DNA molecules and then going on to consider microfibril helices in plant cell walls. Next comes the dextral and sinistral asymmetry of motile cells. If one looks specifically at mosses, peristome twisting and protonemal phototropic and thigmotropic responses immediately come to mind. With the addition of this kind of information this paper could be of general interest but not as it stands at present.
R1. We thank the Reviewer for raising this relevant issue. As suggested, we have integrated Section 1 (‘The asymmetric nature of the universe’; Lines 36-54, p. 2) in the new version of the manuscript. The following references have been added:
1) Inaki, M.; Liu, J.; Matsuno, K. Cell chirality: its origin and roles in left–right asymmetric development. Philos. Trans. R. Soc. Lond. B Biol. Sci., 2016, 371, 20150403. https://doi.org/10.1098/rstb.2015.0403
2) Taniguchi, K.; Maeda, R.; Ando, T.; Okumura, T.; Nakazawa, N.; Hatori, R.; Nakamura, M.; Hozumi, S.; Fujiwara, H.; Matsuno, K. Chirality in planar cell shape contributes to left-right asymmetric epithelial morphogenesis. Science, 2011, 333, 339-341. DOI: 10.1126/science.1200940
3) Lengyel, J. A.; Iwaki, D. D. It takes guts: the Drosophila hindgut as a model system for organogenesis. Dev. Biol., 2002, 243, 1–19. doi:10.1006/dbio.2002.0577
4) Yamanaka, H.; Kondo, S. Rotating pigment cells exhibit an intrinsic chirality. Genes Cells, 2015, 20, 29–35. doi:10.1111/gtc.12194
5) Naganathan, S. R.; Fürthauer, S.; Nishikawa, M.; Jülicher, F.; Grill, S. W. Active torque generation by the actomyosin cell cortex drives left–right symmetry breaking. eLife, 2014, 3, e04165. doi:10.7554/eLife.04165
6) Wan, L. Q.; Ronaldson, K.; Park, M.; Taylor, G.; Zhang, Y.; Gimble, J. M.; Vunjak-Novakovic, G. Micropatterned mammalian cells exhibit phenotype-specific left–right asymmetry. Proc. Natl Acad. Sci. USA, 2011, 108, 12 295–12 300. doi:10.1073/pnas.1103834108
7) Chen, T. H.; Hsu, J. J.; Zhao, X.; Guo, C.; Wong, M. N.; Huang, Y.; Li, Z.; Garfinkel, A.; Ho, C. M.; Tintut, Y.; Demer, L.L. Left–right symmetry breaking in tissue morphogenesis via cytoskeletal mechanics novelty and significance. Circul. Res., 2012, 110, 551–559. doi:10.1161/CIRCRESAHA.111.255927
Reviewer 4 Report
Comments and Suggestions for Authors
I appreciate the invitation to review this paper. My expertise resides in behavioral genetics in animal models and the development of statistical approaches to behavioral assessment. The authors present a review on how handedness in animals may be also relevant and feasible to be studied in plants. The paper presented was very intriguing in what it proposes. I must note the paper reads more like a perspective than a review. I strongly suggest having this paper reclassified as a perspective paper.
I have several concerns with the paper. First, the concept of “handedness” is associated to behavior and preference, a concept well defined in humans that has been translated to some degree to animal examples. This concept is not well defined in plants, “handedness” is therefore a problematic word to use in this narrative.
Second, the main argument on the paper is that it does not propose any common etiology to animals. Etiology is never included or argued in the paper, and this is problematic considering the previous point of defining behavior and preference in plants. For example, the paper mentions the asymmetry of organ distribution in humans that is very consistent with the heart being most often on the left side and the liver most often on the right side. The paper also mentions the strong 90% predominance of right handedness in humans. However, these two phenomena are not necessarily related. Left-handed people have their heath and liver most often in the same sides as right-handed people. Therefore, how does the main argument on handedness and morphological asymmetry relate to their etiology. To me, they don’t.
Third, effects such as heliotropism are never mentioned. Following the argument starting in line 208 on examples of plants that have a preferred direction while also mentioning the hemisphere where these plants grow. From the fixed perspective of a plant, the sun always travels in the same direction during the day. This creates phenomena such as wood grain and spiral growth effects, these effects are influenced by latitude effects. A question I have on this phenomenon is: How do plants that evolved in one hemisphere grow in the opposite hemisphere? This makes the idea of the genetic basis of asymmetry very compelling. Are these phenomena relevant when discussing climbing plants starting in lines 351. These plants evolved more likely in only one hemisphere.
I would like to invite the authors to make an argument on my critique and clarify their narrative in their paper. If this is not possible, I cannot recommend this paper for publication in the journal.
Author Response
1. First, the concept of “handedness” is associated to behavior and preference, a concept well defined in humans that has been translated to some degree to animal examples. This concept is not well defined in plants, “handedness” is therefore a problematic word to use in this narrative.
R1. We thank the Reviewer for raising this issue. We have integrated the text of the new version of the manuscript by providing a proper definition of handedness in plants (lines 156-163 p. 5; 244-246, p. 7).
2. Second, the main argument on the paper is that it does not propose any common etiology to animals. Etiology is never included or argued in the paper, and this is problematic considering the previous point of defining behavior and preference in plants. For example, the paper mentions the asymmetry of organ distribution in humans that is very consistent with the heart being most often on the left side and the liver most often on the right side. The paper also mentions the strong 90% predominance of right handedness in humans. However, these two phenomena are not necessarily related. Left-handed people have their heath and liver most often in the same sides as right-handed people. Therefore, how does the main argument on handedness and morphological asymmetry relate to their etiology. To me, they don’t.
R2. We agree and we added some specific consideration on this in the conclusion section (lines 481-493, p. 13).
3. Third, effects such as heliotropism are never mentioned. Following the argument starting in line 208 on examples of plants that have a preferred direction while also mentioning the hemisphere where these plants grow. From the fixed perspective of a plant, the sun always travels in the same direction during the day. This creates phenomena such as wood grain and spiral growth effects, these effects are influenced by latitude effects. A question I have on this phenomenon is: How do plants that evolved in one hemisphere grow in the opposite hemisphere? This makes the idea of the genetic basis of asymmetry very compelling. Are these phenomena relevant when discussing climbing plants starting in lines 351. These plants evolved more likely in only one hemisphere. I would like to invite the authors to make an argument on my critique and clarify their narrative in their paper. If this is not possible, I cannot recommend this paper for publication in the journal.
R3. We thank the Reviewer for bringing our attention to this point. We have now integrated the text following the reviewer’s suggestion (lines 264-269, p. 8; lines 283-286, p. 8; 297-306 p. 8). The following references have been added:
1) Bose, J. C.; Guha, S. C. The dia-heliotropic attitude of leaves as determined by transmitted nervous excitation. Proceedings of the Royal Society of London. Series B, Containing Papers of a Biological Character, 1922, 93, 153-178. https://doi.org/10.1098/rspb.1922.0011
2) Atamian, H. S.; Creux, N. M.; Brown, R. I.; Garner, A. G.; Blackman, B. K.; Harmer, S. L. Circadian regulation of sunflower heliotropism, floral orientation, and pollinator visits. Science, 2016, 353, 587-590. Doi:10.1126/science.aaf9793
3) Souza, G. M.; Viana, J. D. O. F.; Oliveira, R. F. D. Asymmetrical leaves induced by water deficit show asymmetric photosynthesis in common bean. Braz. J. Plant Physiol, 2005,17, 223-227. https://doi.org/10.1590/S1677-04202005000200005
4) Lüttge, U.; Souza, G. M. The Golden Section and beauty in nature: The perfection of symmetry and the charm of asymmetry. Prog. Biophys. Mol. Bio., 2019, 146, 98-103. https://doi.org/10.1016/j.pbiomolbio.2018.12.008
Round 2
Reviewer 2 Report
Comments and Suggestions for Authors
The authors have improved their manuscript but, of course, they cannot completely address my reservations about drawing commonalities between laterality in plants and animals. Nevertheless, their attempt to draw the readers’ attention to a possible association between laterality in plants and animals makes interesting reading and is, perhaps, timely.
11: I do not think that ‘brainless’ is the best word here. It would be better stated as ‘organisms without a nervous system such as plants”.
17: The information inside the brackets is too broad. Therefore, I suggest changing this to sentence to ‘..and some lateralities are homologous’..
19: Change ‘The most prominent consequence…’ to ‘The most well-known expression of…’.
20: Delete ‘in animals’ since handedness in non-human species is not common, as the authors make clear in their paper. Furthermore, the Abstract needs to be altered to remove the word ‘handedness’ and so match the now revised text to follow.
58-60: Is there any evidence that ‘animals with frontal rather than lateral eyes divide their perceptual world along coordinates such as right-left, top-bottom and front-back’? If so, please cite it. Also, is this ability known only for animals with frontal eyes or could it even be so for species with lateral eyes?
114-118: References need to be cited to support this evidence. Also, say where asymmetry of the organs is associated in any way with asymmetry in the brain.
196: Correct to ‘handedness in humans appears’.
202: It would be better as ‘…, a laterality which is unmatched by limb preference in other species,…’.
206-208: This sentence is incorrect. Although limb preferences have been reported in these species there is no evidence of it as population-level asymmetry. There is convincing evidence for population-level asymmetry in non-human species but that is for eye/brain asymmetries, not for limb preference.
219-220: This final sentence of the paragraph is logically impossible/incorrect. A study cannot be ‘freed from the presence of limbs’. I see what the authors aim to point out but this sentence needs to be corrected.
313: Correct ‘refer’ to ‘refers’.
458: Insert ‘states’ before that.
469: Delete the comma before ‘can’.
482: Delete ‘both’.
484: Insert an apostrophe in ‘plants’ (possessive case).
Comments on the Quality of English Language
The English needs some corrections, as I have listed above.
Author Response
1. 11: I do not think that ‘brainless’ is the best word here. It would be better stated as ‘organisms without a nervous system such as plants”.
R1. We have now reworded the sentence in the last version of the manuscript. Please refer to p. 1 line 11.
2. 17: The information inside the brackets is too broad. Therefore, I suggest changing this to sentence to ‘..and some lateralities are homologous’..
R2. We have now reworded the sentence in the last version of the manuscript. Please refer to p. 1 lines 16-17.
3. 19: Change ‘The most prominent consequence…’ to ‘The most well-known expression of…’.
R3. We have now reworded the sentence in the last version of the manuscript. Please refer to p. 1 line 19.
4. 20: Delete ‘in animals’ since handedness in non-human species is not common, as the authors make clear in their paper. Furthermore, the Abstract needs to be altered to remove the word ‘handedness’ and so match the now revised text to follow.
R4. We have deleted the word ‘animals’ as suggested by the Reviewer. Furthermore, we do get the point and when possible, we have replaced the term “handedness”. Yet as stated in the manuscript we highlighted that we are using the term handedness with caution referring to plants and we stressed the undeniable difference between humans, animals and plants.
5. 58-60: Is there any evidence that ‘animals with frontal rather than lateral eyes divide their perceptual world along coordinates such as right-left, top-bottom and front-back’? If so, please cite it. Also, is this ability known only for animals with frontal eyes or could it even be so for species with lateral eyes?
R5. We have reworded the sentence to make it clearer and we have added a reference in support (lines 58-61, p.2)
Voss, J., & Bischof, H. J. (2009). Eye movements of laterally eyed birds are not independent. Journal of Experimental Biology, 212(10), 1568-1575.
6. 114-118: References need to be cited to support this evidence. Also, say where asymmetry of the organs is associated in any way with asymmetry in the brain.
R6. As reported in Barth et al 2015 visceral, neuroanatomical and behavioural asymmetries are linked in fsi zebrafish. We added this evidence and a further reference.
7. 196: Correct to ‘handedness in humans appears’.
R7. We have reworded the sentence in the last version of the manuscript. Please refer to p. 6 line 197.
8. 202: It would be better as ‘…, a laterality which is unmatched by limb preference in other species,…’.
R8. We have reworded the sentence in the last version of the manuscript. Please refer to p. 6 lines 203-204.
9. 206-208: This sentence is incorrect. Although limb preferences have been reported in these species there is no evidence of it as population-level asymmetry. There is convincing evidence for population-level asymmetry in non-human species but that is for eye/brain asymmetries, not for limb preference.
R9. We agree with the Reviewer. We have now corrected the sentence accordingly.
10. 219-220: This final sentence of the paragraph is logically impossible/incorrect. A study cannot be ‘freed from the presence of limbs’. I see what the authors aim to point out but this sentence needs to be corrected.
R10. We have reworded the sentence in the last version of the manuscript. See p. 6, lines 220-221.
11. 313: Correct ‘refer’ to ‘refers’.
R11. Done.
12. 458: Insert ‘states’ before that.
R12. Done.
13. 469: Delete the comma before ‘can’.
R13. Done
14. 482: Delete ‘both’.
R14. Done.
15. 484: Insert an apostrophe in ‘plants’ (possessive case).
R15. Done.
Reviewer 3 Report
Comments and Suggestions for Authors
The revised version is much improved and now starts with an appropriate wide-ranging introduction. The text is a little verbose and awkward and would be improved with light editing. Overall however, this is an interesting synthesis with novel comparisons between animals and plants.
Comments on the Quality of English LanguageNeeds light editing
Author Response
1. The revised version is much improved and now starts with an appropriate wide-ranging introduction. The text is a little verbose and awkward and would be improved with light editing. Overall however, this is an interesting synthesis with novel comparisons between animals and plants.
R1. We thank the Reviewer for his/her suggestions. The manuscript has been edited by a native speaker to improve the quality of the English language.